# Characteristics and Mechanism of Crayfish Myofibril Protein Gel Deterioration Induced by Autoclaving

**DOI:** 10.3390/foods11070929

**Published:** 2022-03-23

**Authors:** Xu Kang, Meihu Ma, Jianglan Yuan, Yaming Huang

**Affiliations:** 1College of Food Science and Technology, Huazhong Agricultural University, Wuhan 430068, China; kangxu2000@126.com; 2College of Bioengineering and Food, Hubei University of Technology, Wuhan 430068, China; ymwyyx163@163.com

**Keywords:** crayfish, autoclaving, myofibril protein, gel, deterioration

## Abstract

Crayfish myofibril protein (CMP) gel deterioration induced by autoclaving was investigated. A series of CMP gels were obtained through treating CMP solutions at different autoclaving conditions from 100 °C/0.1 MPa to 121 °C/0.21 MPa, and then characteristics and the mechanism of gel texture deterioration along with the intensification of autoclaving were explored through determining appearance, texture, protein composition, cross-linking forces, degree of hydrolysis, water state, microstructure of the gels, and average particle size of aggregates. When autoclaving was at above 105 °C/0.103 MPa, texture of CMP gel showed a tendency to severely weaken with the intensification of autoclaving (*p* < 0.05), hydrophobic interaction and aggregation between proteins weakened gradually (*p* < 0.05), and moderately bound water in the gel decreased and T_22_ relaxation time significantly increased (*p* < 0.05). After heating for 30 min at above 105 °C/0.103 MPa, pores in the microstructure of CMP gel enlarged obviously, and myosin heavy chain (MHC) degraded. It can be concluded that CMP gel deterioration induced by autoclaving was associated with the degradation of MHC and 105 °C might be the critical temperature to ensure good texture of crayfish products.

## 1. Introduction

*Procambarus clarkia* (known as crayfish) is native to northeastern Mexico and south-central America [1]. In recent years, it has been widely artificially farmed in China and become a common freshwater economy species in inland waters of China due to its delicious taste and rich nutrition. In season, crayfish is mainly processed into a popular and delicious ready-to-eat dishes. Moreover, crayfish are also processed into some products for cross-seasonal and cross-regional consumption [2]. However, the texture of crayfish decreases with improper heat processing, which could have something to do with crayfish meat protein gel.

Gel is very important for sensory qualities of meat products due to its abilities to bind water, stabilize fat, and so on. Myofibril protein is the main protein in animal muscle, and plays a critical role in texture of meat and aquatic products through producing a three-dimensional viscoelastic gel network [3]. Like myofibril protein in artiodactyl meat, crayfish myofibril proteins (CMP) consist of salt-soluble proteins in the muscle, including myosin heavy chain (MHC, 200 kDa; myosin light chain (MLC, 20 kDa), actin (45 kDa), paramyosin (100 kDa), tropomyosin (38 kDa), troponin T, and so on [4], which is also particularly important for crayfish products due to their heat-induced gelation characteristics.

Heating is an essential operation for obtaining gel-type meat products, and so achieving a commercially desirable texture [5]. In food processing, autoclaving is a common heating method to achieve the purpose of cooking and sterilizing, and in the meantime, the proteins can form a heat-induced gel. Compared with the other proteins in meat, it is easier for myofibril protein to form a heat-induced gel due to its relatively low critical concentration of gelation [6]. It was reported that MHC and actin were mainly responsible for heat-induced gelation in fish surimi [7,8], and it has been proved that high temperature processing can lead to degradation of myofibril protein, leading to gel destruction in fish products [9]. However, there are few studies on gel of arthropod crustacean meat products.

In the work, the characteristics and mechanism of CMP gel deterioration through autoclaving were investigated. The research may provide theoretical guidance for the further research and processing of crayfish meat.

## 2. Materials and Methods

### 2.1. Materials

Fresh crayfish (*P. clarkii*) was purchased from the local market. Maleic acid (HPLC), ortho-phthalaldehyde (OPA, purity ≥ 98%) and DL-Serine (≥98%) from Macklin Co., Ltd. (Shanghai, China); N,N,N’,N’-Tetramethylethylenediamine (TEMED), Tris base (≥99.9%), β-mercaptoethanol (≥99%), and Coomassie brilliant blue R-250 from Sigma-Aldrich (St. Louis, MO, USA); acrylamide (≥99.5%), N,N-methylene bisacrylamide (≥99%), and Sodium dodecyl sulfate (SDS) from BioSharp Co., Ltd. (Shanghai, China); glycine and protein standards for SDS-PAGE from TAKARA Co., Ltd. (Tokyo, Japan); DL-dithiothreitol (DTT, purity ≥ 99%) and Coomassie brilliant blue G-250 from Fei Yang Biotechnology Co., Ltd. (Xi’an, China). All other reagents were of analytical grade.

### 2.2. Extraction of CMP

CMP was extracted from fresh crayfish according to the method in the references [10,11]. Meat was peeled off from tail of live crayfish, and then mixed with 10 times the volume of buffer A (20 mM Tris-maleic acid solution containing 0.05 M sodium chloride, pH 7.0), homogenized for 1 min at 4 °C with a food processor, immediately centrifuged at 10,000 rpm for 15 min at 4 °C, the precipitation was collected and washed 3 times by 5 times volume of buffer A. The precipitation was mixed with 10 times volume of buffer B (20 mM Tris-maleic acid solution containing 0.6 M sodium chloride, pH 7.0), followed by centrifuging at the same conditions, and then the supernatant was collected. Finally, the supernatant was mixed with 10 times volume of cold deionized water, centrifuged at 8000 rpm for 20 min at 4 °C, then the precipitation was collected and washed 3 times with cold deionized water, finally, lyophilized for subsequent experiments.

### 2.3. Autoclaving Treatment of CMP

CMP was dissolved in 0.6 M NaCl solution and stirred gently for 12 h at room temperature to obtain 4.0% (*w*/*v*) solution with fully hydrated protein, and then 10.0 mL of the solution was added into a glass petri dish with a diameter of 35 mm, incubated for 1 h in a water bath of 40 °C, and then treated for 30 min in an autoclave at 100 °C/0.10 MPa, 105 °C/0.103 MPa, 110 °C/0.15 MPa, 115 °C/0.175 MPa, 118 °C/0.20 MPa, and 121 °C/0.21 MPa, respectively. Finally, the samples were immediately cooled to room temperature, and refrigerated for later analysis.

### 2.4. Texture Analysis

Hardness and springiness of CMP gels were determined by using a TA.XT Plus texture analyzer (Stable Micro Systems, Godalming, UK) with a cylinder probe (*p*/0.5 R, 0.5 inch in diameter) according to the method described by Chen et al. [12]. CMP gels were cut into cuboid with uniform sizes (10 × 10 × 5 mm^3^). Compression degree was 50%, pre- and post-test speed was set at 1 mm/s, and test speed was set at 0.5 mm/s. Trigger type was set as “auto”, and trigger force was 5.0 g.

### 2.5. Sodium Dodecyl Sulfate–Polyacrylamide Gel Electrophoresis (SDS-PAGE)

SDS-PAGE was performed according to the method described in the reference [13]. The gels (prepared in Section 2.3) were broken into pieces and then centrifuged for 5 min at 10,000 rpm to separate the solution from the matrix. The matrices were washed with ultrapure water and centrifuged for 5 min at 10,000 rpm, repeated for three times, and finally, grinded well as the sample for electrophoresis. Protein composition in the solution and the matrix was determined by SDS-PAGE. The samples were mixed with loading buffer of equal volume, and then boiled for 3 min, followed by centrifuging for 2 min at 10,000 rpm. Finally, 10 µL of the supernatant was loaded into the well. The concentrations of spacer and separation gel were 4% and 12%, respectively.

### 2.6. Degree of Hydrolysis

Degree of hydrolysis (DH) was determined by OPA method described by Nielsen et al. [14] with slight modifications. OPA reagent was prepared as follows: 9.525 g sodium borate and 250 mg SDS were dissolved in deionized water (called A1); 200 mg OPA was dissolved in 5 mL anhydrous ethanol (A2); 220 mg DTT was dissolved in a small amount of deionized water (A3). A2 and A3 were transferred quantitatively to A1 by rinsing with deionized water, and finally, the mixed solution was diluted to 250 mL with deionized water to obtain OPA reagent. The serine standard was prepared as follows: 50 mg serine was dissolved in 500 mL deionized water (serine-NH_2_ 0.9516 meqv/L). The sample solution (1.0 mg/mL) was prepared as follows: 10 mg freeze-dried powders of CMP gels and CMP were dissolved in 10 mL 0.6 mol/L NaCl solution.

DH of protein in the gels was quantified by serine standard. The standard was measured as follows: 400 µL serine standard was mixed with 3 mL OPA reagents in a test tube, and absorption value at 340 nm was immediately measured by an UV–Vis spectrophotometer (TU1900, Persee Co., Beijing, China) after mixing for 5 s and standing for 2 min, recorded as *A_st_*. All samples were measured as described above and recorded as *A_sa_*. Then, 400 μL deionized water was applied as blank and measured as described above, and recorded as *A*_0_. DH of the sample was calculated according to following formulas.
$${\mathrm{SerineNH}}_{2}=\frac{A_{sa}-A_{0}}{A_{st}-A_{0}}\times 0.9516\;\mathrm{meqv}/L$$
$$h=\frac{{\mathrm{SerineNH}}_{2}-\beta }{\alpha }\mathrm{meqv}/g\;\mathrm{protein}$$
$$\mathrm{DH}=\frac{h_{1}-h_{0}}{h_{tot}}\times 100\%$$
where serineNH_2_ means how much meqv serineNH_2_ is contained in a gram of protein; α and *β* for fish are 1.00 and 0.40, respectively; *h*_0_ and *h*_1_ is the degree of hydrolysis of the sample compared with original substrate before and after hydrolysis reaction respectively; *h_tot_* for fish is 8.6.

### 2.7. Particle Size Analysis

Particle size was analyzed by using a Zetasizer Nano-ZS (Malven Instruments Ltd., Malvern, UK) [12]. The samples were prepared as follows: 10 mg CMP was dissolved in 10 mL 0.6 mol/L NaCl solution, and then autoclaved as described in Section 2.3. Finally, the samples stood overnight at 4 °C. The refractive indexes were 1.330 and 1.450 for the solvent and protein, respectively. Measurements were performed at 25 °C.

### 2.8. Gelation Forces Analysis

Gelation forces were analyzed according to the method reported by Gómez-Guillén et al. [15] with a minor modification. Five kinds of solutions were prepared, including 0.05 M NaCl (SA), 0.6 M NaCl (SB), 0.6 M NaCl + 1.5 M urea (SC), 0.6 M NaCl + 8 M urea (SD), and 0.6 M NaCl + 8 M urea + 1.5 M β-mercaptoethanol (SE). Two grams of chopped gel were homogenized for 2 min with 10 mL of each solution by a vortex mixer, and then stirred at 4 °C for 1 h, centrifuged at 10,000 rpm for 15 min. Protein concentration in supernatants was determined by using Coomassie brilliant blue G250. The gelation forces are calculated and expressed as follows:$$\mathrm{Ionic}\;\mathrm{bonds}\;\left(\mathrm{mg}/\mathrm{mL}\right)=C_{SB}-C_{SA}$$
$$\mathrm{Hydrogen}\;\mathrm{bonds}\;\left(\mathrm{mg}/\mathrm{mL}\right)=C_{SC}-C_{SB}$$
$$\mathrm{Hydrophobic}\;\mathrm{interactions}\;\left(\mathrm{mg}/\mathrm{mL}\right)=C_{SD}-C_{SC}$$
$$\mathrm{Disulfide}\;\mathrm{bonds}\;\left(\mathrm{mg}/\mathrm{mL}\right)=C_{SE}-C_{SD}$$
where *C* means concentration of protein solubilized in each solution (mg/mL).

### 2.9. Low Field Nuclear Magnetic Resonance Measurements

An NMI-20-Analyst NMR analyzer (Niumag Co., Ltd., Shanghai, China) was used to detect water information in gels at molecular level. Transverse relaxation (T_2_) curve was measured by Carr–Purcell–Meiboom–Gill (CPMG) pulse sequence at 32 °C. One gram of gel were wrapped in waterproof film and then placed in NMR test tube (15 mm in diameter). Measuring parameters included 5000 ms of recycle time, 1000 of echo count, 8 times of scan repetitions, and 0.7 ms of echo time. Data were analyzed by an inversion to obtain T_2_ spectrum. Each sample was in triplicate.

### 2.10. Scanning Electron Microscopy (SEM)

Microstructure of CMP gel was determined by a scanning electron microscope (SU 8000, Hitachi, Japan) according to the references [16,17] with some modification. The gels were cut into cuboids (10 × 10 × 3 mm^3^), and then submerged in phosphate buffer (0.1 M, pH7.2) with 3.0% glutaraldehyde for 24 h at 4 °C, followed by soaking in phosphate buffer (0.1 M, pH7.2) for 30 min so as to wash off glutaraldehyde. The fixed gel blocks were dehydrated by a graded series of ethanol (30%, 50%, 70%, 90%, and 100%) for approximately 15 min for each step, and immediately blew for 5 min with cool air to remove residual organic reagents. The dehydrated gel blocks were frosted by liquid nitrogen, and then freeze-dried for 24 h. Finally, fracture surfaces of gel blocks were sputter-coated with gold by a carbon coater prior to observation. Accelerating voltage was 3 kV and magnification was 10,000×.

### 2.11. Statistical Analysis

Statistical analysis was performed using Origin 9.0 and SPSS 19.0 software package. Each experiment was repeated three times at least, except as otherwise noted. Data were expressed as means ± standard deviation. Significance of difference between data was analyzed by Duncan test, and level was defined at *p* < 0.05 throughout the study.

## 3. Results and Discussion

### 3.1. Texture of CMP Gel

Appearance and texture of CMP gels were shown in Figure 1A,B. When temperature/pressure of autoclaving was no more than 105 °C/0.103 MPa, surfaces of the gels were smooth and compact, gel blocks appeared good uniformity, integrity, and elasticity. However, when temperature/pressure was above 105 °C/0.103 MPa, surfaces of the gels appeared defective and inhomogeneous, and became weaker and weaker along with the intensification of autoclaving. When temperature/pressure reached up to 121 °C/0.21 MPa, the gel collapsed, and gel characteristics basically disappeared.

Myofibril protein is the main component of meat in domestic animals and responsible for the gelation of meat products, so changes in its structure and function during processing could alter the texture of gel-type meat products and regulate releasing of myofibrillar water [5,18,19]. Similarly, the main protein in crustacean meat is also myofibril protein, which is susceptible to denaturation, aggregation, and cross-linkage by heating, and forms heat-induced gel and endows corresponding products with good texture [20] thus, CMP gel was closely related to texture of cooked crayfish.

The results from Figure 1B show that the springiness of CMP gels declined constantly with enhancement of autoclaving, while hardness and springiness decreased rapidly when temperature/pressure of autoclaving was over 105 °C/0.103 MPa. Therefore, it is obvious that autoclaving can cause CMP gel deterioration at above 105 °C. On the one hand, autoclaving can cause thermal denaturation of CMP, and then promote subsequent gelation of CMP. However, the structure of the gel can be also destroyed through strengthening autoclaving, reducing the strength and water holding capacity of the gel, and increasing the juice loss of crayfish gel-type meat products.

### 3.2. Change in Proteins in CMP Gels

The change in proteins was investigated by using SDS-PAGE to analyze the inclusion solution (Figure 2A) and matrix (Figure 2B) in CMP gel. The results in Figure 2A show that centrifugal solution from CMP gel obtained by the various autoclaving was mainly composed of tropomyosin and MLC. However, some new bands clearly appeared when temperature/pressure of the autoclaving reached 115 °C/0.175 MPa, and meanwhile tropomyosin faded away; however, MLC had no obvious change, indicating that tropomyosin tended to degrade through autoclaving. Moreover, some proteins in the gel matrixes could be severely degraded and transferred from gel matrix to the solution, and so resulted in some new bands.

It can be seen from Figure 2B that the main proteins in gel matrix were MHC, actin, and tropomyosin. The band of MHC weakened gradually along with the intensification of autoclaving, and began to disappear at 115 °C/0.175 MPa, suggested that MHC was almost completely degraded, and transferred from matrix to inclusion solution. The gels were completely dependent on cross-linking of actin aggregates under the conditions. Therefore, actin was also involved in the formation of CMP gel, and presented a relatively good stability but weak cross-linking ability. MHC was more susceptible to degrade than actin. Although MHC and actin were the main proteins to form CMP gel network, MHC was primarily responsible for the deterioration of CMP gel. The conclusion can be drawn that the deterioration of CMP gel was adversely affected by autoclaving due to the obvious degradation of MHC above 110 °C.

Like other animal meat [7,9,21], MHC and actin were the main proteins in crayfish meat and are responsible for the texture of the products by constructing a gel network. Under the synergistic effect of the pressure, 110 °C might be a critical temperature value for significant degradation of MHC and tropomyosin, leading directly to deterioration of CMP gel, as can be confirmed from Figure 1. Proteolytic degradation changed the appearance and texture of CMP gel, followed by serious water loss. Therefore, moderate autoclaving can be seen to promote crayfish meat to form a good texture due to denaturation, aggregation, and the cross-linking of meat proteins including MHC, actin, and tropomyosin, whereas excessive autoclaving destroys the structure of the gel-type products mainly by degrading MHC. Autoclaving below 105 °C/0.103 MPa is reliable for crayfish meat.

The effect of autoclaving on DH of CMP is presented in Figure 2C. DH of CMP increased gradually from 1.41% to 10.43% when autoclaving increased from 100 °C/0.10 MPa to 121 °C/0.21 MPa. High-temperature processing results in degradation of animal muscle proteins. Beef proteins degraded during high temperature processing, and DH was correlated positively with temperature and time. At the initial stage, beef myofibril protein formed a gel, but this was followed by the obvious degradation of MHC, which eventually degraded into peptides and free amino acids [22]. Similar phenomena have also been reported in fish, chicken, and pork [23,24,25]. Therefore, it was speculated that the degradation of CMP (mainly MHC) resulted in texture deterioration of crayfish meat due to gel weakening.

### 3.3. Aggregation and Cross-Linking Forces of CMP

Heat-induced gelation usually goes through three stages including denaturation, aggregation, and crosslinking of protein molecules, and so the particle size of protein aggregates in the CMP solution treated by autoclaving is helpful to clarify the forming and weakening trend of CMP gel, and the results are shown in Figure 3A. Average particle size of unheated CMP solution was about 500 nm, but they aggregated into particles of about 1600 nm after being treated for 30 min at 100 °C/0.10 MPa. However, the size of protein aggregates decreased along with the strengthening of autoclaving due to degradation of CMP. It had been reported that larger protein aggregates were beneficial to form a gel network with excellent WHC and springiness in protein heat-induced gel [26]. The results in Figure 3A suggest that it was increasingly difficult for CMP to aggregate along with the intensification of autoclaving, and accordingly, the gelation ability of CMP reduced. Degradation of CMP could account for the phenomenon.

Cross-linking forces in gel structure are directly related to texture of the gel. It has been proved that the forces involved in the formation of myofibril protein gel mainly include hydrogen bonds, electrostatic interaction, hydrophobic interaction, and some covalent bonds (mainly disulfide bond) [27]. In fact, formation of protein gel resulted from the balance of various chemical forces. The results in Figure 3B indicate that hydrophobic interaction played a crucial role in the process of CMP gelation. Forces of the gels weakened gradually with the enhancement of autoclaving, among which hydrophobic interaction and ionic bonds decreased significantly. Gelation of CMP mainly depended on hydrophobic interaction because a large number of hydrophobic groups were exposed to the surface of protein molecule after autoclaving, and thereby surface hydrophobicity of the molecules increased, and furthermore aggregation between proteins occurred due to the hydrophobic interaction. However, hydrophobic interaction decreased significantly above 110 °C/0.15 MPa probably due to protein degradation [28]. A slight upward trend of disulfide bonds can also be observed along with the enhancement of autoclaving, which are related to the exposure and oxidation of sulfhydryl [22]. Autoclaving can cause degradation of CMP, and so results in deterioration of the gels by weakening the aggregation and cross-linking forces between proteins, and finally, adversely affecting the quality of crayfish meat.

### 3.4. Low Field Nuclear Magnetic Resonance (LF-NMR) Analysis

LF-NMR is an accurate and efficient method to determine the state and distribution of water molecules according to their mobility. T_2_ relaxation times indicate bound degree of water molecule. The peaks T_21_, T_22_, and T_23_ were shown in the T_2_ spectrum of the gel samples by various autoclaving (Figure 4), The peak T_21_ represented the tightly bound water with macromolecules, the peak T_22_ was attributed to moderately bound water entrapped within enclosed spaces in gel network, and the peak T_23_ corresponded to the free water that was bound loosely and fluidly in the gel network [29,30].

Peak area was related to water content, as shown in Table 1. The results from Table 1 suggest that only a small percentage of water (all below 1.0%) in the gels was bound with CMP molecules or their aggregates, and no significant difference between the samples was detected for P_21_ (*p* < 0.05). The peak area P_22_ suggests that moderately bound water was the most abundant in CMP gels, and that water molecules were stranded in the gel network. However, it could be observed that moderately bound water decreased along with the enhancement of autoclaving, indicated that the gel network was destroyed gradually and so the space of entrapping water reduced. Meanwhile, free water increased along with the decrease in moderately bound water. Obviously, moderately bound water in the gel treated by 100 °C/0.10 MPa was the most abundant, and the relaxation time T_22_ was gradually prolonged with the increase in autoclaving, and accordingly, both P_23_ and T_23_ of free water increased gradually. Overall, the water-holding capacity of CMP gel decreased significantly along with strengthening of autoclaving.

### 3.5. Microstructure of CMP Gel

Research on the gel microstructure could be helpful to understand its properties [31]. Microstructures of CMP gels after various autoclaving are presented in Figure 5. A compact gel network was formed by autoclaving below 105 °C/0.103 MPa. Along with the intensification of autoclaving, the network of CMP gels became looser and looser, and pores became larger and larger, suggested that cross-linking degree of gel gradually reduced. Denatured myosin could be served as active gel building blocks as it is highly reactive [32], while degradation of MHC accounted for the alteration in the gel microstructure at above 105 °C. Therefore, a negative correlation of autoclaving on the microstructure of CMP gel can observed, especially at above 105 °C, which was consistent with the results above.

## 4. Conclusions

The present study explored the effect of autoclaving on the deterioration of CMP gel, and thereby revealed the characteristics and mechanism of degradation in the quality of crayfish products processed at high temperature and high pressure. Appropriate autoclaving could promote CMP to form gel, and endow cooked crayfish with excellent texture. However, excessive autoclaving had an obvious unfavorable impact on the texture of CMP gel, particularly above 105 °C/0.103 MPa. Degradation of MHC at autoclaving could be a key factor causing deterioration of CMP gel. Degradation of CMP above 105 °C/0.103 MPa provides a full explanation for the deterioration trend of CMP gel, and further explains the degradation in the quality of crayfish products by autoclaving. The results can provide a theoretical basis for crayfish processing.

## Figures and Tables

**Figure 1 foods-11-00929-f001:**
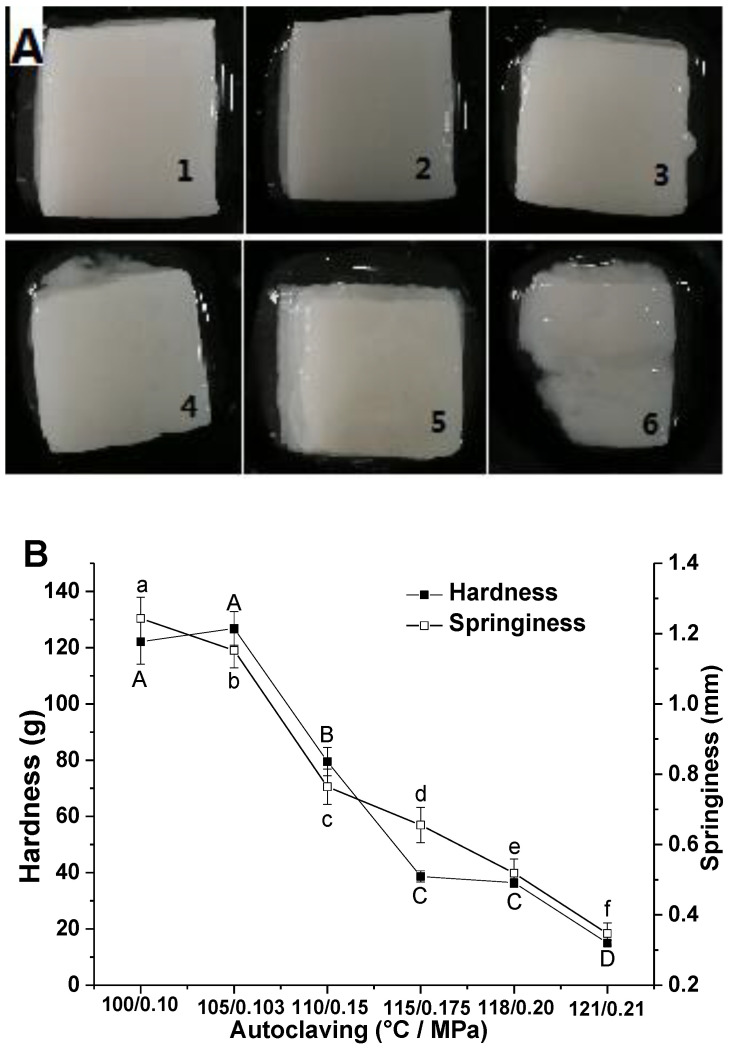
Effect of autoclaving on appearance (**A**) and texture (**B**) of CMP heat-induced gel. The gels from 1–6 in A are obtained by treating CMP solutions at 100 °C/0.1 MPa, 105 °C/0.103 MPa, 110 °C/0.15 MPa, 115 °C/0.175 MPa, 118 °C/0.20 MPa and 121 °C/0.21 MPa, respectively. In B, different letters indicate significant difference in hardness (uppercase) and springiness (lowercase) between different samples (*p* < 0.05). Bars indicate standard deviations of the means (*n* ≥ 3).

**Figure 2 foods-11-00929-f002:**
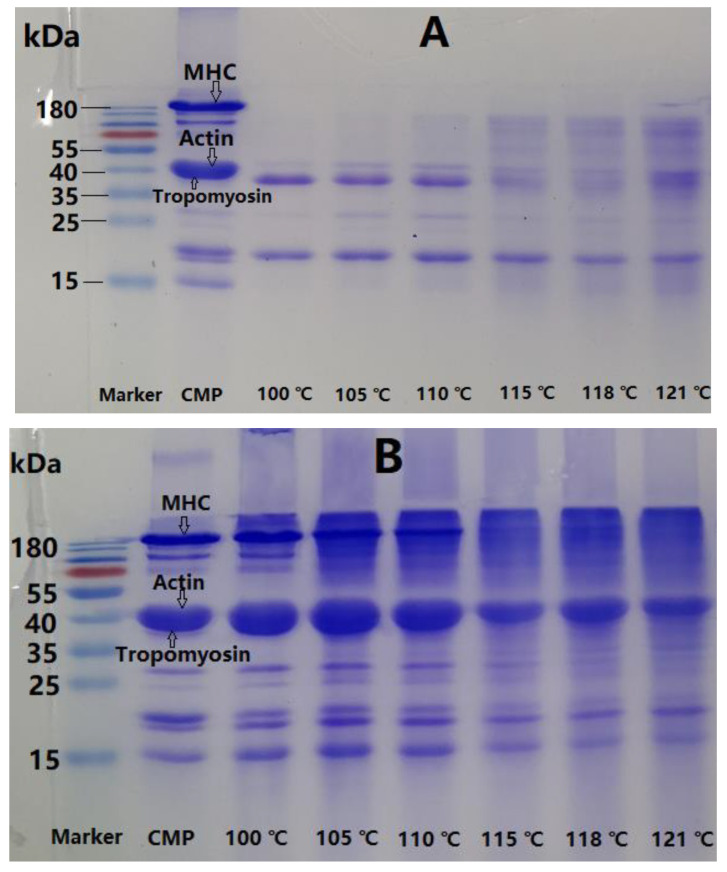
Change in proteins in CMP gel after various autoclaving: (**A**) SDS-PAGE of centrifugal solutions from various CMP gels; (**B**) SDS-PAGE of various CMP gel matrixes; (**C**) Degree of CMP hydrolysis, the letters from “a”–“f” indicates the significance of DH difference after various autoclaving (*p* < 0.05).

**Figure 3 foods-11-00929-f003:**
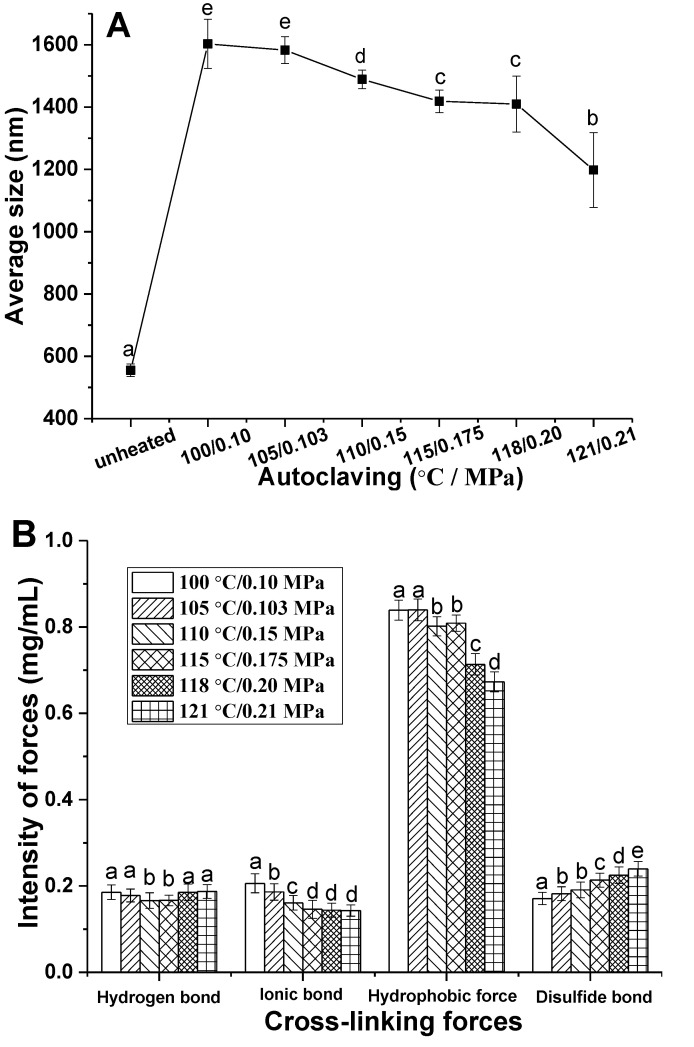
(**A**) Average size of aggregates in CMP solutions after various autoclaving; (**B**) Cross-linking forces in various CMP gel. Bars indicate the standard deviations of the means (*n* = 3), and different lowercase letters (a–e) on the top of the columns represent significant difference in the same cross-linking force between CMP gels obtained at various autoclaving according to Duncan tests (*p* < 0.05).

**Figure 4 foods-11-00929-f004:**
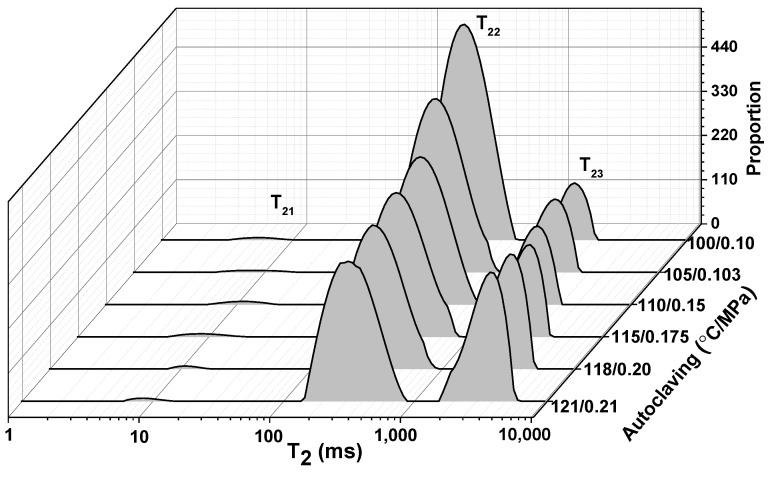
LF-NMR T_2_ relaxation spectra of the gel samples prepared with crayfish myofibrillar protein (CMP) by different autoclaving conditions.

**Figure 5 foods-11-00929-f005:**
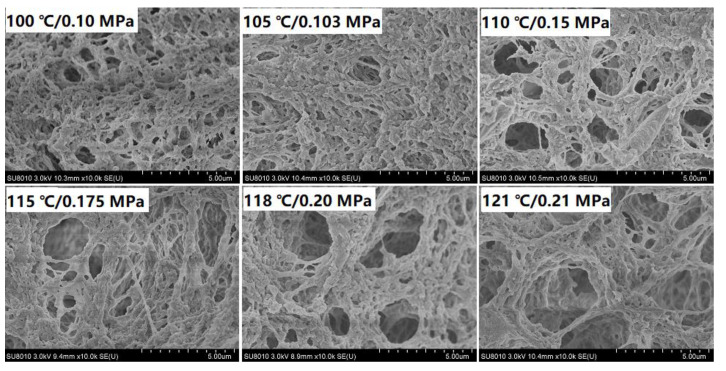
SEM images (10,000×) of CMP heat-induced gel obtained by various AT.

**Table 1 foods-11-00929-t001:** Transverse relaxation time and proportion of peak area.

Treatment	T_21_ (ms)	P_21_ (%)	T_22_ (ms)	P_22_ (%)	T_23_ (ms)	P_23_ (%)
100 ± 0.1 °C	5.5 ± 0.3 ^a^	0.8 ± 0.0 ^a^	209.8 ± 8.5 ^a^	89.7 ± 1.3 ^a^	1367.5 ± 1.7 ^a^	9.7 ± 1.3 ^a^
105 ± 0.1 °C	7.8 ± 0.3 ^b^	0.9 ± 0.0 ^a^	214.7 ± 8.5 ^ab^	75.4 ± 0.2 ^bc^	1399.8 ± 2.3 ^ab^	23.9 ± 0.1 ^b^
110 ± 0.1 °C	11.1 ± 0.4 ^d^	0.9 ± 0.0 ^a^	252.4 ± 9.1 ^b^	74.2 ± 0.3 ^c^	1431.5 ± 0.2 ^b^	24.9 ± 0.3 ^bc^
115 ± 0.1 °C	9.0 ± 0.3 ^bc^	0.9 ± 0.0 ^a^	266.4 ± 9.8 ^b^	73.0 ± 2.1 ^cd^	1438.8 ± 6.0 ^b^	26.2 ± 2.1 ^cd^
118 ± 0.1 °C	11.1 ± 0.4 ^d^	0.7 ± 0.0 ^a^	289.9 ± 0.0 ^c^	65.1 ± 0.9 ^d^	1544.7 ± 12.8 ^c^	33.9 ± 0.8 ^d^
121 ± 0.1 °C	9.7 ± 0.4 ^c^	0.8 ± 0.0 ^a^	310.8 ± 0.0 ^c^	62.2 ± 0.1 ^e^	1602.2 ± 11.5 ^d^	37.0 ± 0.2 ^e^

Lowercase letters (a–e) indicate significant differences (*p* < 0.05) among the samples by different autoclaving. P_21_, P_22_ and P_23_ indicate the relative content of tightly bound water, moderately bound water, and loosely bound water respectively.

## Data Availability

Data is contained within the article.

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
