# Peer review of "Characteristics and Mechanism of Crayfish Myofibril Protein Gel Deterioration Induced by Autoclaving"

_foods, 2022, doi:10.3390/foods11070929_

Round 1

Reviewer 1 Report

The study aimed to evaluate the effect of autoclave heating [oC/MPa: 100/0.10, 105/0.103, 110/0.15, 115/0.175, 118/0.20, and 121/0.21] on the physicochemical [texture (hardness/springiness), particle size, SEM, hydrolysis degree, gelation force] and functional (SDS-PAGE, low-field magnetic resonance) characteristics of Crayfish myofibril protein gels. As expected, several characteristics were modified by thermal treatments, some of them in a dose-dependent manner. Minor changes are suggested to increase the manuscript´s scientific soundness: 

General

  • The manuscript´s reading and comprehension will improve if it is reviewed by a native English-speaking colleague or if it is sent to a formal translation agency.
  • Please reduce as much as possible unneeded abbreviations and put the meaning of those needed the first time they are mentioned.

Abstract. It should be described in a more quantitative way (including p-values) and, if additional statistical analyzes are included, describe changes in a temperature-trend dependent manner.

Methods. Given that various response variables resulting from a dose-dependent experimental design (increase in temperature/pressure) were evaluated, the authors should consider including other statistical tests (in addition to the ANOVA tests) that allow visualizing this trend and the possible correlation ( correlation matrix) between variables.

Results & Discussion. Having decided to carry out the additional statistical tests, it is suggested to discuss each section of results in a more integrative and inductive way (e.g., which variables have high correlation and why?)

Figures. The resolution and sharpness of all figures should improve. It is suggested to put each graph accompanied by its corresponding gel (below), in a larger size.

Table 1. It is advisable to use just one decimal and to include linear regression data (m, R2) to evidence temperature-trend modifications to each parameter.

Author Response

Dear the Reviewer:

Thank you very much for your comments on our manuscript entitled “Characteristics and mechanism of crayfish myofibril protein gel deterioration induced by autoclaving” (Manuscript ID: foods-1608462). The comments are all valuable and very helpful for revising and improving our manuscript, as well as important guiding significance to our research. We have studied the comments carefully and made correction which we hope meet with approval. All changes are marked in red, but deleted contents don’t display.

The main corrections in the revised manuscript and the responses to the Reviewer 1 are as follows (the red part).

Reviewer 1

The study aimed to evaluate the effect of autoclave heating [oC/MPa: 100/0.10, 105/0.103, 110/0.15, 115/0.175, 118/0.20, and 121/0.21] on the physicochemical [texture (hardness/springiness), particle size, SEM, hydrolysis degree, gelation force] and functional (SDS-PAGE, low-field magnetic resonance) characteristics of Crayfish myofibril protein gels. As expected, several characteristics were modified by thermal treatments, some of them in a dose-dependent manner. Minor changes are suggested to increase the manuscript´s scientific soundness:

General

The manuscript´s reading and comprehension will improve if it is reviewed by a native English-speaking colleague or if it is sent to a formal translation agency.

Please reduce as much as possible unneeded abbreviations and put the meaning of those needed the first time they are mentioned.

Response: We are very sorry for the confusions caused by English editing. We have carefully examined and corrected English expression. We hope the English editing in the revised manuscript to meet the requirements of scientific paper. In order to read easily, we reduced some abbreviations including “MP” and “AT”, and for those needed abbreviations, we had put the meaning in the first time they are mentioned.

Abstract. It should be described in a more quantitative way (including p-values) and, if additional statistical analyzes are included, describe changes in a temperature-trend dependent manner.

Response: We agree with the comment. We have rewritten the abstract to meet the requirements of quantification. Duncan test alone was applied to analyze the significance of difference between data, we have tried to describe the changes in a temperature-trend dependent manner.

Methods. Given that various response variables resulting from a dose-dependent experimental design (increase in temperature/pressure) were evaluated, the authors should consider including other statistical tests (in addition to the ANOVA tests) that allow visualizing this trend and the possible correlation (correlation matrix) between variables.

Response: Thank you very much for the comment. It can be found from the results that there is a negative correlation between temperature/pressure and the texture of CMP heat-induced gel, but the accurate dose-dependent relationship cannot be obtained, which should require more extensive and precise experiment. We hope to improve the experimental design in future studies to clarify an accurate dose-dependent relationship.

Results & Discussion. Having decided to carry out the additional statistical tests, it is suggested to discuss each section of results in a more integrative and inductive way (e.g., which variables have high correlation and why?)

Response: Thank you very much for the comment. Because temperature and pressure varied at the same time when autoclaving, it was not clear which variable had more significant influence on the gel deterioration. Accurate correlation between variables and gel texture could be confirmed by a new experimental design such as variable temperature with constant pressure, or variable pressure with constant temperature.

Figures. The resolution and sharpness of all figures should improve. It is suggested to put each graph accompanied by its corresponding gel (below), in a larger size.

Response: We have reedited and enlarged all the figures, and readjusted the text in the figures to improve the resolution and sharpness of the figures.

Table 1. It is advisable to use just one decimal and to include linear regression data (m, R2) to evidence temperature-trend modifications to each parameter.

Response: We have readjusted all the data to keep one decimal. In Table 1, each parameter showed some variation trend, but not linear, so we couldn’t perform linear regression analysis on the results.

Special thanks to the Reviewer 1 for above comments and the opportunity to reconsider the manuscript.

We tried our best to improve the manuscript and made some changes in the manuscript. The changes will not influence the contents and framework of the manuscript.

We appreciate for the Reviewers’ warm work earnestly, and hope that the correction will meet with approval.

Once again, thank you very much for your comments.

Best regards,

Xu Kang, Meihu Ma, Jianglan Yuan, Yaming Huang

Reviewer 2 Report

In the present study, the authors study the autoclaving effect in Crayfish myofibril protein gel deterioration in different temperatures. This is a very interesting contribution for the food and nutrition human, the other hand this work presents some corrections according to the following comments:

Comments to Authors:

  • Page 3, line 98: Part (2.5. SDS-PAGE) needs more detail
  • Page 5, Figure 1A: identify the gels numbers (1-6).
  • Page 6, Figure 2A: By which method did you identify the proteins Myosin, Actin and Tropomyosin at the SDS-PAGE gel.
  • Page 6, Figure 2A and 2B: Corrected “Tropomysin” by “Tropomyosin”
  • Page 6, Figure 2C: Corrected “Temperature” at the x-axis
  • Page 7, Figure 3A; a scale of the abscissa axes is not respected.
  • In the title, the author’s signal “Characteristics and mechanism”, but the paper does not present a mechanism, I suggest eliminating it.

Author Response

Dear the Reviewer 2:

Thank you very much for your comments on our manuscript entitled “Characteristics and mechanism of crayfish myofibril protein gel deterioration induced by autoclaving” (Manuscript ID: foods-1608462). The comments are all valuable and very helpful for revising and improving our manuscript, as well as important guiding significance to our research. We have studied the comments carefully and made correction which we hope meet with approval. All changes are marked in red, but deleted contents don’t display.

The main corrections in the revised manuscript and the responses to the Reviewer 2 are as follows (the red part).

Reviewer 2

In the present study, the authors study the autoclaving effect in Crayfish myofibril protein gel deterioration in different temperatures. This is a very interesting contribution for the food and nutrition human, the other hand this work presents some corrections according to the following comments:

Comments to Authors:

Page 3, line 98: Part (2.5. SDS-PAGE) needs more detail

Response: Thanks for the comments. We have added some details about sample processing.

Page 5, Figure 1A: identify the gels numbers (1-6).

Response: We are so sorry for our carelessness. We have identified the gels numbers (1-6) in Figure 1A

Page 6, Figure 2A: By which method did you identify the proteins Myosin, Actin and Tropomyosin at the SDS-PAGE gel.

Response: We identified these protein bands according to abundance and molecular weights of main proteins in myofibril protein reported in the references, which have been described in introduction of the manuscript.

Page 6, Figure 2A and 2B: Corrected “Tropomysin” by “Tropomyosin”

Response: We are so sorry for the error. We have reedited Figure 2A and 2B, and corrected “Tropomysin” by “Tropomyosin”.

Page 6, Figure 2C: Corrected “Temperature” at the x-axis

Response: We have corrected the x-axis title “Temperature” by “Autoclaving” in Figure 2C, Figure 3A and Figure 4.

In the title, the author’s signal “Characteristics and mechanism”, but the paper does not present a mechanism, I suggest eliminating it.

Response: Thanks for the comment. We had preliminarily explored the mechanism of CMP gel deterioration, and the corresponding experiment were presented in Figure 2 and Figure 3. Accordingly, CMP gel deterioration along with increasing of AT was mainly caused by degradation of MHC, and thereby caused the decreasing of hydrophobic forces in the gel.

Special thanks to the Reviewer 2 for above comments and the opportunity to reconsider the manuscript.

We tried our best to improve the manuscript and made some changes in the manuscript. The changes will not influence the contents and framework of the manuscript.

We appreciate for the Editors and Reviewer’ warm work earnestly, and hope that the correction will meet with approval.

Once again, thank you very much for your comments.

Best regards,

Xu Kang, Meihu Ma, Jianglan Yuan, Yaming Huang

Round 2

Reviewer 2 Report

The authors answered all the questions and made all the corrections requested. I recommend the publication of the paper